# Monitoring of the Dry-Curing Process in Iberian Ham Through the Evaluation of Fat Volatile Organic Compounds by Gas Chromatography–Ion Mobility Spectrometry and Non-Destructive Sampling

**DOI:** 10.3390/foods14010049

**Published:** 2024-12-27

**Authors:** Pablo Rodríguez-Hernández, Andrés Martín-Gómez, Miriam Rivero-Talavera, María José Cardador, Vicente Rodríguez-Estévez, Lourdes Arce

**Affiliations:** 1Departamento de Producción Animal, Facultad de Veterinaria, Universidad de Córdoba, 14071 Córdoba, Spain; pablo.rodriguez.hernandez@uco.es (P.R.-H.); pa2roesv@uco.es (V.R.-E.); 2Departamento de Química Analítica, Instituto de Química para la Energía y el Medio Ambiente, Anexo Marie Curie, Universidad de Córdoba, 14071 Córdoba, Spain; q02magoa@uco.es (A.M.-G.); q72ritam@uco.es (M.R.-T.); qa1arjil@uco.es (L.A.)

**Keywords:** dry-cured Iberian ham, HS-GC-IMS, ripening stages, volatilome analysis: volatile organic compounds

## Abstract

The current quality control of the dry-curing process in Iberian ham is performed with an olfactory evaluation by ham experts. The present study proposes to monitor the dry-curing process of Iberian ham using an objective analytical methodology that involves non-destructive sampling of the subcutaneous fat of the hams and a volatile profile analysis using gas chromatography–ion mobility spectrometry. Thirty-eight 100% Iberian acorn-fed hams were examined in total, with eighteen hams monitored during the post-salting stage and twenty during the drying-maturation stage. A total of 164 markers were detected in the samples monitored during the post-salting stage, whereas 276 were detected in the hams monitored during the drying-maturation stage. The study of the trends observed in the intensities of the markers throughout the curing process enabled the detection of potential indicators of the end of the post-salting stage. Thus, representative intensity thresholds were established for some markers (3-methylbutanal, pentanal, hexanal, (*E*)-octen-2-al, 2-hexen-1-ol and heptan-1-ol) to determine the optimal point to specify the end of the post-salting process for hams. These findings provide an objective tool to support the traditional sensory evaluation currently performed in the industry.

## 1. Introduction

Dry-cured Iberian ham is a foodstuff that is highly appreciated worldwide because of its quality, added value, and traditional production procedure [1]. Iberian ham is a distinctive product with significant economic importance in Spain and Portugal. The production procedure starts at farms with the food given to the pigs during a final fattening phase before sacrifice. Depending on the commercial categories established by regulation [2], it may be an intensive diet based on feed, or an extensive feeding regime during montanera (meaning pannage and foraging acorns) for Iberian acorn-fed pigs. After the pigs are slaughtered, the hams are carefully selected and trimmed. They undergo a salting phase to enhance the flavor and ensure preservation, followed by a post-salting stage where the salt penetrates uniformly. Finally, they are aged in natural and artificial cellars during the drying-maturation stage, allowing the development of their characteristic flavor and aroma. The whole procedure can take up to 5 years, finishing with a complex and long dry-curing process which lasts over 24–36 months in the case of the acorn-fed ham, the most valued product coming from this breed [3]. Meticulous control is required in each stage of the elaboration process. There are many factors and variables which have a substantial influence on the final characteristics of the product.

During the dry-curing of hams, different enzymatic and biochemical changes take place and contribute to their aroma, taste, and texture. These changes affect different components, such as the water and salt content, nitrogenated compounds, and lipids. Among these, reactions affecting lipids are highlighted due to their important influence on ham aroma. Hydrolytic lipolysis releases fatty acids, which afterwards serve as substratum for oxidative processes and provide volatile organic compounds (VOCs) [4,5,6,7]. The complex collection of VOCs resulting from these changes and reactions constitutes Iberian ham’s aroma, an important quality attribute.

In this regard, the dry-cured Iberian ham aroma is evaluated to determine quality aspects, such as curing status or sensory defects. Aroma evaluation is currently performed in the industry by using a traditional olfactory methodology called cala (meaning puncture). This is performed by a trained expert (maestro jamonero) and consists of inserting a bone awl into different parts of hams (e.g., the rump). Then, this expert, by means of smell, evaluates the quality of each ham and whether it is free of defects. However, this sensory evaluation is conditioned by the subjectivity and fatigue of the master ham expert. Therefore, there is a growing interest in the industry in the development of analytical methodologies to evaluate the quality of ham and help in decision making [8].

Over the years, many studies have been carried out using analytical techniques to monitor the dry-curing process of Iberian hams. Antequera et al. (1992) [9] studied the evolution of acidity, the peroxide value, and the aldehydes content in order to assess the extent of lipid oxidation. However, the methodology employed was based on a destructive sampling of the ham due to the need to perform sample pretreatments, such as solid–liquid extraction [9]. Other authors have used analytical methods based on gas chromatography–mass spectrometry (GC-MS) to study the evolution of a large number of chemical compounds, such as fatty acids, aldehydes, ketones, hydrocarbons, alcohols, and esters [10,11,12,13,14] obtaining trends that may be useful for monitoring the dry-curing of hams. These methodologies were based on a destructive sampling to obtain samples that were subjected to solid-phase microextraction (SPME) or the dynamic headspace technique for the determination of polar compounds and VOCs [10,11,12,13,14]. Headspace gas chromatography–ion mobility spectrometry (HS-GC-IMS) combined with a destructive sample collection methodology has been also used to study the evolution of some VOCs from traditional Chinese dry-cured ham (Jinhua) [15] to identify a total of 53 VOCs in 7 curing stages. Some of them were highlighted (butanal, 3-methylbutanal, 2-methylbutanal, 2-hexanone, 2-pentanone, and 2-butanone) as significant markers in aroma evolution in this type of dry-cured ham.

Despite the interesting findings underlined by the above-mentioned studies, the sample collection methodologies used have important drawbacks for the industry. As an example, destructive sampling of hams impedes their sale as entire pieces. Therefore, sampled hams have to be sliced to be sold. In light of this situation, Martín-Gómez et al. (2019) [16] proposed a non-destructive sampling method based on ham punction with a sterile disposable needle. Once the needle is impregnated, VOCs are extracted by directly heating the needle in a headspace vial and analyzed by HS-GC-IMS. This sampling protocol may be very useful for the industry to evaluate ham aroma as it does not alter the ham’s integrity and can be used at any point along the distribution chain. In fact, it has already been used in previous studies for VOC evaluation from Iberian hams pursuing different objectives, such as the authentication of the feeding regime supplied to Iberian pigs [16,17], the origin discrimination of Iberian ham [18], and the identification of defective hams [8].

The main goal of this study was to monitor the dry-curing process of Iberian hams by tracking the evolution of markers detected with HS-GC–IMS technique. This study provides a novel contribution in monitoring the curing process to find markers that indicate the end of each stage in the curing process, providing an objective tool to support sensory analysis in the industry. In service of that aim, two independent batches of hams were included: one batch was monitored through the post-salting stage, while the other batch was analyzed during the drying-maturation stage. Each batch was sampled and analyzed over several months to study the evolution of VOCs in each curing stage and to detect potential markers which may provide interesting information about the process. The samples analyzed consist of subcutaneous fat extracted from the hams using a non-destructive sampling method optimized by Martín-Gómez et al. [16], which allows the same ham to be consistently evaluated throughout the curing process.

## 2. Materials and Methods

### 2.1. Samples and Standards

38 100% Iberian acorn-fed hams were selected for the present study: 18 were sampled during the post-salting stage, and the remaining 20 were sampled in a natural cellar during the drying-maturation stage. Two different batches of hams were used for post-salting and drying-maturation stages to avoid a four-year wait to complete the study. Both batches were handled in the same way during processing in the industry. Figure 1 represents the sampling times for the hams at each stage of the curing process. The post-salting stage was monitored over a period of five months, with a total of four samplings. The drying-maturation stage was monitored during the last 18 months of the process in 14 samplings. Sampling was conducted every six weeks to capture the curing process evolution at intervals enough to detect significant changes in the markers, without causing data redundancy.

The hams included in the study were provided by a ham-curing plant registered under the Protected Designation of Origin of Los Pedroches (Córdoba, Spain), and sample collection was carried at its facilities. The non-destructive sampling method previously described [16] was performed using sterile 2.1 × 60 mm disposable stainless-steel needles (Bovivet-Kruuse, Langeskov, Denmark) to puncture the rump area of the ham, allowing the needle to become impregnated with the subcutaneous fat of the ham. The metal part of the needle was cut with pliers and placed in a 20 mL headspace glass vial which was subsequently closed with an aluminum cap and a silicone septum. The needles impregnated with subcutaneous fat were stored at −18 °C for five to seven days maximum until undergoing laboratory analysis in order to prevent lipid oxidation.

A ketone mix of 0.5 mg/L was prepared by dissolving six high purity (≥99%) ketones (nonan-2-one, octan-2-one, heptan-2-one, hexan-2-one, pentan-2-one, and butan-2-one) (Sigma Aldrich, Madrid, Spain) in ultrapure water (Milli-Q Plus, Millipore Bedford, MA, USA) as a quality control for the instrument to guarantee accuracy throughout the study. Every day, 1 mL of the ketone mix solution was analyzed before each sample tray. The reactant ion peak (RIP) intensity and information regarding each ketone (intensity, retention, time and drift time) were monitored.

### 2.2. Instrumentation and Method

The instrumentation used (HS-GC-IMS) consisted of an Agilent 7697A HS sampler, which was coupled via a transfer line to an Agilent 8860 gas chromatograph (Agilent, Santa Clara, CA, USA) and to a standalone ion mobility spectrometer (G.A.S. Gesellschaft für analytische Sensorsysteme mph, Dortmund, Germany) with a ^3^H ionization source and a 10 cm drift tube.

The filling pressure and pressure equilibration time in the HS sampler were set at 14 psi and 1 min, respectively. Every impregnated needle was heated in the HS oven at 60 °C for 15 min to promote the extraction of VOCs to the HS of the vial. Subsequent, HS from the vial was collected through a 1 mL sample loop (100 °C), then transferred to the GC via a heated transfer line (110 °C), and finally injected in the injection port of the GC (100 °C). The injection was performed in split mode (1:5 split ratio) for 0.5 min. The GC integrated a 30 m nonpolar HP-5 (5%-phenyl-95% methylpolysiloxane) column with an internal diameter of 0.32 mm and a 0.5 µm film thickness (Agilent, Santa Clara, CA, USA). The initial GC oven temperature was 40 °C for 3 min, increased first to 100 °C at a rate of 5 °C/min, then to 130 °C at a rate of 15 °C/min, and ended at a plateau of 130 °C until the end of the process (total runtime: 27 min). After separation on the GC column, the VOCs entered the ionization chamber of the IMS module, whose detector was operated with positive polarity. The IMS parameters were set as follows: 150 µs injection pulse width, 45 °C drift tube temperature, signal averaging every 32 spectra, a repetition rate of 30 ms, and drift, blocking and injection voltages of 237 V, 40 V and 2500 V, respectively. Helium was used as the carrier gas at a constant flow rate of 1 mL/min and nitrogen was used as the drift gas at a flow of 150 mL/min.

### 2.3. Data Processing and Statistical Analysis

Data processing was performed independently for the two batches of hams evaluated, namely the 18 hams sampled during the post-salting stage and the 20 hams monitored in the course of drying-maturation stage. The approach used for both data treatments was the same.

First, VOCal 1.0.0. software (G.A.S. Gesellschaft für analytische Sensorsysteme mbH, Dortmund, Germany) was used for GC-IMS data exportation. LAV 2.2.1. software (G.A.S.) was subsequently used to visualize the data, align the topographic maps, and obtain the individual maximum volumes of the signals. The tentative identification of these signals in the topographic maps was performed by comparing their retention index (RI) and IMS drift time (DT) with the standards in the GC × IMS library integrated in the VOCal software (GAS, Dortmund, Germany). This software library contains a comprehensive database of RI and DT for a wide range of volatile compounds, allowing for more accurate identification through compound pattern similarity analysis (fingerprint). Firstly, the data were normalized using the ketone mix standard (0.5 mg/L) to calculate the RI of the compounds and to support compound identification via retention index libraries (NIST2014 RI database included in the VOCal software). Comparison through the RI and the DT ensures a more accurate identification and helps reduce potential misidentifications.

The intensity of the markers in the topographic maps were calculated as the maximum volume of the signal. No quantitative analysis was performed in order to simplify the methodology, making it more practical for potential industry implementation. OriginPro^®^ 2024 software was used for the calculation of Pearson correlation coefficients (r), which were used to measure the linear relationship between the intensity of the markers in the samples and the ripening time. The value of r allowed to establish an increasing/positive (r > 0.5), decreasing/negative (r < −0.5), or constant trend (−0.5 < r < 0.5) in the signal intensity [19]. Absolute r values of > 0.8 were considered as very strong positive or negative correlations between signal intensity and ripening time. The interval 0.8 > r > 0.7 was considered as a strong positive or negative correlation. The interval 0.7 > r > 0.6 indicated a moderate positive or negative correlation. The interval 0.6 > r > 0.5 indicated a weak positive or negative correlation. Finally, absolute r values < 0.5 indicated no correlation whatsoever between signal intensity and ripening time, i.e., a constant trend [19].

## 3. Results and Discussion

Table 1 shows the VOCs tentatively identified using the GC-IMS library and their evolution during the stages evaluated, including their RI, DT, Spearman correlation r values, *p* values, and the type of trend. It is essential to take into account that in IMS, the ionization of a single compound though protonation may produce two or more distinct ionic species; in this case, monomers and dimers with different DT were formed. Most of the compounds tentatively identified in this study had been previously reported in dry-cured Iberian hams by several authors [11,13,14,20,21,22,23,24], with the exception of 2,5-dimethylpirazine.

### 3.1. VOCs Profile in Post-Salting and in Drying-Maturation Stages

The visual evaluation of the GC-IMS topographic maps of the post-salting samples revealed a total of 164 markers (signals), whereas 276 were detected in the samples from the drying-maturation stage. The greater number of markers detected in the last stage is consistent with the large number of VOCs generated during the curing process, which gradually increases until the formation of a mature ham flavor [15,25]. Among the 164 markers detected in the post-salting stage, 56 ions, corresponding to 36 compounds, were tentatively identified based on the GC-IMS library: 12 aldehydes, 8 alcohols, 5 ketones, 1 ester, 5 carboxylic acids, and 5 aromatic compounds (Table 1). On the other hand, 47 ions, corresponding to 27 compounds were tentatively identified in the drying-maturation stage: 13 aldehydes, 4 alcohols, 6 ketones, 2 carboxylic acids, and 2 aromatic compounds (Table 1). Therefore, the contents of aldehydes, alcohols, and ketones are highlighted as the most abundant groups in both post-salting and in drying-maturation stages, which is in line with previous studies where these three families were the major groups of compounds detected [23]. This same abundance distribution of aldehydes, alcohols, and ketones was also found in a study about flavor formation in Jinhua dry-cured ham, also using HS-GC-IMS [15]. Nevertheless, it should be mentioned that most of the previous studies about dry-cured Iberian ham used SPME-GC-MS, while Narváez-Rivas et al. [23] reported that the volatile profile of cured Iberian ham may exhibit slight differences in detection depending on the technique employed.

Apart from identification of VOCs, Table 1 also shows the intensity trend during the stages evaluated and the Spearman correlation coefficients (r) to measure the strength of the relationship between signal intensity and ripening time [19]. The signal intensities of most of the compounds identified throughout the post-salting stage (23 out of 36, understood as the sum of the intensities of their product ions) had an increasing trend, showing from weak to very strong positive Spearman correlations (from 0.502 to 0.897). Only 1 VOC showed a decreasing trend in the post-salting stage (isovaleric, with r −0.631) while 11 showed a constant trend (nonanal, propan-1-ol, acetic acid ethyl ester, 2-ethylfuran, 1-hydroxypropan-2-one, 3-hydroxybutan-2-one, hexan-2-one, acetic acid, propanoic acid, 2-methylbutanoic acid, and hexanoic acid). Of these, isovaleric acid, propan-1-ol, acetic acid ethyl ester, 1-hydroxypropan-2-one, 3-hydroxybutan-2-one, 2-methylbutanoic acid, and hexanoic acid were not subsequently detected in the drying-maturation stage, which is a meaningful finding, since most of them have been related to spoilage in dry-cured hams [25,26]. It is noteworthy that certain compounds exhibited an important initial intensity increase during the post-salting, followed by a subsequently decrease or stabilization towards the drying-maturation stage (3-methylbutanal, pentanal, hexanal, heptanal, (*E*)-hepten-2-al, (*E*)-octen-2-al, pentan-1-ol, heptan-1-ol, 1-octen-3-ol or octan-1-ol) or were even undetected afterwards (2-hexen-1-ol, aniline or 2,5-dimethylpyrazine). This finding is consistent with previous studies evaluating dry-cured Iberian ham, where several aldehydes firstly exhibited an increase, due to their formation by means of lipid oxidation, followed by a subsequent decrease. This was probably due to further chemical reactions of aldehydes with other components, promoted by the high temperatures reached in summer during the drying-maturation stage [21,27]. Due to the seasonality of the slaughter of acorn-fed Iberian pigs (in January–February), the beginning drying-maturation stage coincides with the summer, taking advantage of the natural temperature to complete this stage in cellars.

With respect to the drying-maturation stage, most of the VOCs had constant trends (21 out of 27), showing Spearman correlations between −0.500 to 0.500. On the other hand, six VOCs showed a decreasing trend ((*E*)-hepten-2-al, (*E*)-octen-2-al, (*E*)-nonen-2-al, (*Z*)-decen-2-al, 1-octen-3-one, and acetic acid) (Table 1). However, those VOCs with decreasing trends mostly exhibited weak Spearman correlations (absolute r values from 0.558 to 0.609). It should be noted that seven VOCs which had not been detected in the post-salting stage emerged in the second monitored period. This may be attributed to their development during the initial maturation months in the cellar, where environmental temperature fluctuations (between 15–25 °C depending on the season) play an important role. Moreover, some studies have highlighted the important role that microorganisms play in the formation of compounds during the cellar maturation [23].

The above-mentioned trends were especially notable in the aldehyde fraction, where 11 of the 12 identified aldehydes increased their intensity during the post-salting stage due to lipid oxidation [28,29] as well as due to protein and Strecker degradation [11]; afterwards, their intensities remained constant or decreased during the drying-maturation stage (Table 1), which may be caused by condensation reactions during this final part of the process [23]. These reactions can occur between the previously formed aldehydes, in addition to reactions between the aldehydes and the large quantity of free amino acids present at these stages, leading to the formation of Maillard compounds [30]. There are some exceptions, such as the evolution of lineal aldehydes, like decanal, originating from lipid autooxidation [15], which was not detected during the post-salting stage and reached its maximum concentration at the end of the ripening process, in the drying-maturation stage. The continuous increase observed for 3-methylbutanal during the post-salting aligns with the results of Jurado et al. (2009) [20], who identified this compound as the most abundant VOC in hams.

Alcohols were the second most abundant family of compounds identified in this study during the two stages evaluated (Table 1). The formation of alcohols in dry-cured ham is generally promoted by the oxidation of fatty acids, such as myristic, oleic, linoleic and palmitoleic acid [15]. However, branched alcohols also come from the Maillard reaction and the Strecker degradation, and they may suffer a further oxidation [11,14,15]. Most of the alcohols detected in this study increased their intensities during the post-salting stage and showed a constant intensity trend during the drying-maturation stage, or even disappeared (Table 1). Only propan-1-ol and 2,3-butanediol showed constant intensities during the post-salting stage.

The mechanism of formation of ketones, in general, is similar to the one for aldehydes, i.e., through the degradation of lipids [28,29]. In fact, ketones have been mainly described as intermediates of lipid oxidation [15,29]. In the present study, these compounds exhibited a constant or moderate increasing trend during the post-salting stage (Table 1). Only the monomer ion associated with 1-hydroxypropan-2-one showed a decreasing trend. It should be also noted that 3-hydroxybutan-2-one and pentane-2,3-dione showed a positive or constant trend at the beginning of the post-salting stage. However, they were not detected in the subsequent maturation. Moreover, the increasing intensity of heptan-2-one during the first stage and its subsequent stabilization is consistent with previous studies of other types of dry-cured hams [21]. During the drying-maturation stage, the majority of compounds showed a constant or decreasing trend.

Only one ester was detected in the sampled hams, namely acetic acid ethyl ester, which showed a constant trend only in the post-salting stage, while disappearing in the drying-maturation stage (Table 1). This compound is probably formed from the esterification of carboxylic acids and alcohols [23], which is consistent with the results obtained for these two families of compounds (Table 1). In fact, none of the carboxylic acids detected during the post-salting stage showed an increasing intensity in the drying-maturation stage, and some of them even disappeared, likely due to their conversion into esters. This would also explain the above-mentioned results for alcohols, which mostly exhibited an increasing trend in post-salting, but they were not detected or showed a stable trend later during the maturation process (Table 1). The formation of carboxylic acids in Iberian ham is not clear. According to some authors, these compounds are originated from carbohydrate fermentation by microorganisms [31] and according to others from the Maillard reaction [32]. However, Ramírez and Cava [33] show that these compounds are mainly generated by reactions of lipid oxidation. In the present study, only propanoic and acetic acids were detected in the drying-maturation stage. These two compounds have also been described in Parma [34], Jinhua [35], and American country [36] cured hams. In the case of Iberian ham, propanoic and acetic acids have previously been identified in spoiled dry-cured Iberian ham [25], although their origin remains unclear. Kandler et al. [34] indicated that acetic acid may originate from carbohydrate fermentation by microorganisms, while Martín et al. [32] associated it with the Maillard reaction.

### 3.2. VOCs as Indicators for the End of the Post-Salting Stage

The 56 markers detected and identified in the post-salting stage were studied individually to evaluate if they could be used as potential indicators of the end of this curing stage. Although the post-salting stage usually concluded around the fourth month (red dashed line in graphics of Figure 2) in the studied facilities under the guidance of the master ham artisan, the present study monitored up to week 21 at four time points (weeks 5, 11, 15, and 21 of this curing stage) to evaluate the intensity trends of the markers after the end of this stage. In this sense, the intensity levels reached by these markers around the 21st week could serve as potential thresholds for advancement to the next stage. Consequently, hams that do not reach the expected VOC intensity would require continued processing in the post-salting stage [37].

Only markers with very strong correlations with a very high statistical significance (*p* < 0.01) can be considered as reliable indicators of the curing process. Thus, out of the 56 markers detected and identified in the post-salting stage, only 6 of them exhibited very strong positive trends (r > 0.8). In this stage, markers with very strong negative trends (r < −0.8) were not detected in this post-salting stage.

Figure 2 shows the average intensity of these 6 markers in the 18 hams evaluated. The post-salting stage traditionally concludes around the fourth month. Therefore, a red dashed line has been added at this point in the figure to potentially estimate intensity thresholds for the markers corresponding to the expected value at the end of the fourth month, which may help to determine the end of the stage. The high dispersion in the data for some markers makes it difficult to establish a reliable intensity threshold, as the threshold established at the end of the post-salting stage may fall within the interval of the previous sampling point, which complicates decision making (Figure 2). This variability among the data can be explained because these hams come from extensive systems where pigs have a natural grazing diet, frequently leading to intake differences between individuals [38]. Additionally, the composition of acorns varies throughout the fattening period [39]. These factors contribute to the intrinsic diversity in the aroma and flavor of acorn-fed Iberian hams. In contrast, representative thresholds for the end of the post-salting stage can be established for 3-methylbutanal, pentanal, hexanal, *(E)*-octen-2-al, 2-hexen-1-ol, and heptan-1-ol at 0.432, 0.492, 1.697, 0.285, 0.294, and 0.060 mV, respectively. In fact, some of these identified compounds have been previously described as being responsible for the aroma of dry-cured Iberian ham [23]. For example, *(E)*-2-octenal is one of the most abundant aldehydes in the process and one of the most significant VOCs in contributing to the dry-cured Iberian ham flavor [23]. Moreover, 3-methylbutanal and hexanal have previously been highlighted as quality markers in dry-cured ham [21].

### 3.3. Evaluation of VOCs as Indicators of the End of Drying-Maturation Stage

During the drying-maturation stage, the absolute Spearman correlations between marker intensity and ripening time were weak at most for all 47 markers detected and identified (0.070 < r < 0.609). Figure 3 shows the trends of the six markers with the highest r values, namely *(E)*-hepten-2-al, 1-octen-3-one, *(E)*-nonen-2-al, *(E)*-octen-2-al, *(Z)*-decen-2-al, and acetic acid, with r values of −0.609, −0.562, −0.549, −0.569, −0.588, and −0.558, respectively. The average intensity of these markers in the studied hams showed a decreasing trend throughout this stage, from ripening month 33 to 50. The high variability in data from the 20 hams under study makes these markers a less reliable indicator of the end of the drying-maturation stage. This variability among the data from different hams is reasonable, given the complexity of reactions occurring during the drying-maturation stage, which contributes to the rich aroma profile and intrinsic diversity of acorn-fed Iberian hams. However, the continuous decreasing trend observed in the intensity of these markers can still provide valuable insights into the progression of the ripening process. Thus, the linear fit for *(E)*-2-heptenal highlights its decreasing trend throughout the drying-maturation stage, which is expected, since this compound has been associated with rancid and unpleasant odors in other meat products, characteristics generally considered undesirable [40,41]. Acetic acid, which fits a polynomial equation with a decreasing trend, is also linked to negative attributes in meat; in fact, it is one of the most abundant acids found in spoiled hams, likely formed due to microbial growth in meat products [25,26,42]. Other unidentified markers were also evaluated; however, their intensity trends were weak. In summary, the trends of *(E)*-hepten-2-al, 1-octen-3-one, *(E)*-nonen-2-al, *(E)*-octen-2-al, *(Z)*-decen-2-al, and acetic acid could help to predict maturation problems in the hams.

## 4. Conclusions

The HS-GC-IMS technique, combined with a non-destructive sampling of the subcutaneous fat of the hams, allowed for effective monitoring the curing process. Non-destructive sampling enabled repeated sampling of the same hams throughout the study period without compromising product integrity or value. The high sensitivity of HS-GC-IMS allowed the detection of a broad range of signals in the topographic maps, highlighting the abundance of VOCs generated during the curing process. By analyzing the intensity trends of each marker, representative thresholds were established for six markers (3-methylbutanal, pentanal, hexanal, *(E)*-octen-2-al, 2-hexen-1-ol, and heptan-1-ol), which can be used as indicators of the end of the post-salting stage.

Finally, studying marker intensity trends during the drying-maturation stage is more complex than in the post-salting stage, as correlations between marker intensity and ripening time are lower (from weak to moderate). Moreover, there is a considerable variability in marker intensities among the 20 hams evaluated, likely due to the increased complexity of reactions occurring during the drying-maturation stage. Therefore, it is difficult to find markers that can be used as reliable indicators of the end of this stage. Because of this, the end of the drying-maturation stage based on other parameters, such as weight or water content. However, the study of the evolution of some markers by HS-GC-IMS can be a useful tool in the industry to detect problems in the maturation process of the hams.

In conclusion, the proposed methodology offers a non-invasive, quick, and straightforward approach to ensure the proper dry-curing of Iberian hams and provides objective data to aid master ham artisans in making informed decisions about the curing process.

## Figures and Tables

**Figure 1 foods-14-00049-f001:**
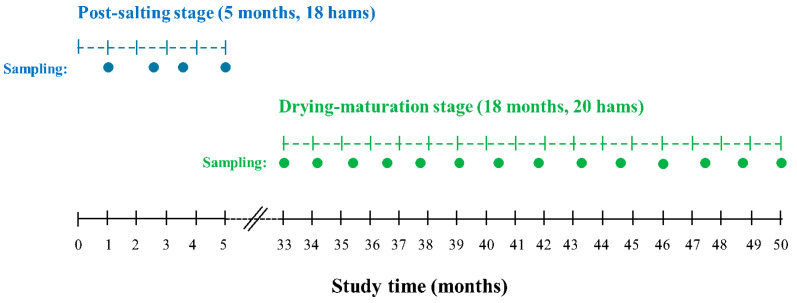
Sampling times during the post-salting (blue dots) and drying-maturation (green dots) stages.

**Figure 2 foods-14-00049-f002:**
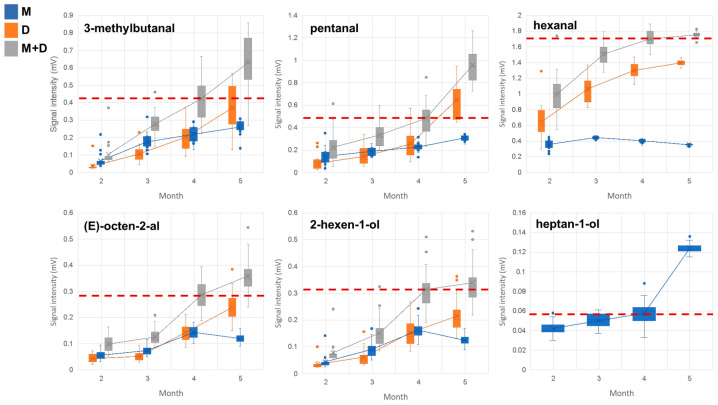
Boxplot with average intensity of the markers with r > 0.8 in the 18 ham samples analyzed in the post-salting stage. The red dashed line represents the expected value of intensity at the end of the post-salting stage (around the fourth month). M: monomer, D: dimer, M+D: sum of the monomer and dimer.

**Figure 3 foods-14-00049-f003:**
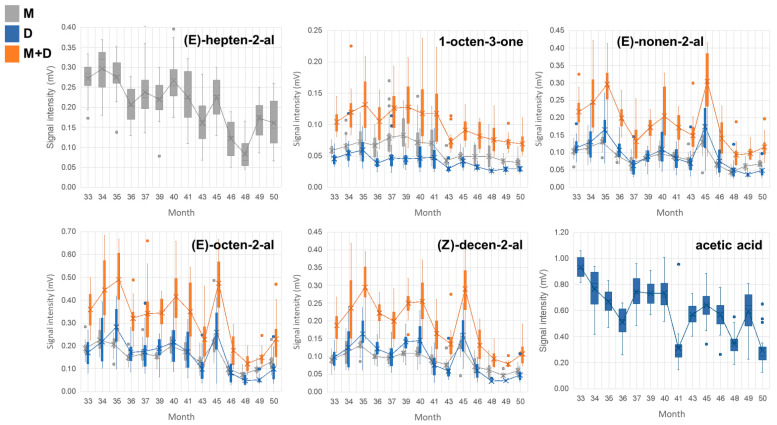
Boxplot with average intensity of the markers with absolute r > 0.6 in the 20 ham samples analyzed in the drying-maturation stage. M: monomer, D: dimer, M+D: sum of the monomer and dimer.

**Table 1 foods-14-00049-t001:** VOCs identified in samples of subcutaneous fat from 100% Iberian acorn-fed hams in both post-salting (18 hams) and drying-maturation (20 hams) stages, as well as their RI, DT, Spearman correlations coefficients (r), *p* values, and intensity trend during the ripening time.

			*Post-Salting*			*Drying-Maturation*		
Compound	RI	DT (ms)	r	*p*	Trend	r	*p*	Trend
*Aldehydes*								
butanal (M)	602	11.2	0.638	***	↑↑	0.240	***	=
3-methylbutanal (M+D)	656	-	0.866	***	↑↑↑↑	−0.291	***	=
3-methylbutanal (M)		11.7	0.830	***	↑↑↑↑	0.233	***	=
3-methylbutanal (D)		14.2	0.878	***	↑↑↑↑	−0.353	***	=
2-methylbutanal (M)	667	11.6	N.F.			−0.070	-	=
pentanal (M+D)	700	-	0.860	***	↑↑↑↑	−0.336	***	=
pentanal (M)		11.9	0.807	***	↑↑↑↑	−0.250	***	=
pentanal (D)		14.4	0.873	***	↑↑↑↑	−0.344	***	=
hexanal (M+D)	805	-	0.826	***	↑↑↑↑	−0.111	-	=
hexanal (M)		12.6	−0.169	-	=	0.258	***	=
hexanal (D)		15.8	0.898	***	↑↑↑↑	−0.145	*	=
heptanal (M+D)	909	-	0.681	***	↑↑	−0.476	***	=
heptanal (M)		13.3	0.502	***	↑	−0.454	***	=
heptanal (D)		17.0	0.739	***	↑↑↑	−0.468	****	=
*(E)*-hepten-2-al (M+D)	966	-	0.796	***	↑↑↑	−0.609	***	↓↓
*(E)*-hepten-2-al (M)		12.5	0.652	***	↑↑	−0.609	***	↓↓
*(E)*-hepten-2-al (D)		16.7	0.856	***	↑↑↑↑	N.F.		
*(E,E)*-heptadien-2,4-al (M+D)	1006	-	0.646	***	↑↑	N.F.		
*(E,E)*-heptadien-2,4-al (M)		12.0	0.265	**	=			
*(E,E)*-heptadien-2,4-al (D)		16.3	0.748	***	↑↑↑			
octanal (M+D)	1012	-	0.638	***	↑↑	−0.287	***	=
octanal (M)		14.1	0.446	***	=	−0.252	***	=
octanal (D)		18.3	0.743	***	↑↑↑	−0.295	***	=
*(E)*-octen-2-al (M+D)	1068	-	0.878	***	↑↑↑↑	−0.569	***	↓
*(E)*-octen-2-al (M)		13.3	0.766	***	↑↑↑	−0.573	***	↓
*(E)*-octen-2-al (D)		18.2	0.893	***	↑↑↑↑	−0.554	***	↓
nonanal (M+D)	1109	-	0.121	-	=	−0.289	***	=
nonanal (M)		14.8	−0.162	-	=	−0.296	***	=
nonanal (D)		19.5	0.319	***	=	−0.273	***	=
*(E)*-nonen-2-al (M+D)	1141	-	0.591	***	↑	−0.549	***	↓
*(E)*-nonen-2-al (M)		14.1	0.164	-	=	−0.549	***	↓
*(E)*-nonen-2-al (D)		19.7	0.727	***	↑↑↑	−0.543	***	↓
*(Z)*-2-decenal (M+D)	1192	-	0.647	***	↑↑	−0.588	***	↓
*(Z)*-2-decenal (M)		12.1	0.481	***	=	−0.526	***	↓
*(Z)*-2-decenal (D)		17.2	0.687	***	↑↑	−0.602	***	↓↓
decanal (M+D)	1206	-	N.F.			−0.265	***	=
decanal (M)		15.4				−0.263	***	=
decanal (D)		20.7				−0.259	***	=
*Alcohols*								
propan-1-ol (M)	538	11.2	0.163	-	=	N.F.		
2-methylbutan-1-ol (M+D)	745	-	0.668	***	↑↑	N.F.		
2-methylbutan-1-ol (M)		12.4	0.735	***	↑↑↑			
2-methylbutan-1-ol (D)		14.6	−0.002	-	=			
pentan-1-ol (M+D)	774	-	0.721	***	↑↑↑	−0.197	***	=
pentan-1-ol (M)		12.6	0.648	***	↑↑	−0.239	***	=
pentan-1-ol (D)		15.1	0.815	***	↑↑↑↑	−0.140	**	=
2,3-butanediol (M)	789	13.9	−0.195	-	=	N.F.		
*(E)*-2-hexen-1-ol (M+D)	860	-	0.855	***	↑↑↑↑	N.F.		
*(E)*-2-hexen-1-ol (M)		11.8	0.708	***	↑↑↑			
*(E)*-2-hexen-1-ol (D)		15.1	0.897	***	↑↑↑↑			
heptan-1-ol (M+D)	982	-	0.820	***	↑↑↑↑	−0.276	***	=
heptan-1-ol (M)		14.0	0.820	***	↑↑↑↑	−0.295	***	=
heptan-1-ol (D)		17.6	N.F.			−0.227	***	=
1-octen-3-ol (M)	989	11.6	0.704	***	↑↑↑	−0.430	***	=
octan-1-ol (M+D)	1083	-	0.741	***	↑↑↑	−0.295	***	=
octan-1-ol (M)		14.7	0.741	***	↑↑↑	−0.319	***	=
octan-1-ol (D)		18.8	N.F.			−0.136	**	=
*Esters*								
acetic acid ethyl ester (M)	601	10.9	0.422	***	=	N.F.		
*Aromatics*								
2-ethylfuran (M+D)	711	-	0.102	-	=	N.F.		
2-ethylfuran (M)		10.5	−0.440	***	=			
2-ethylfuran (D)		13.1	0.692	***	↑↑			
aniline (M+D)	807	-	0.744	***	↑↑↑	N.F.		
aniline (M)		11.6	−0.227	-	=			
aniline (D)		14.2	0.892	***	↑↑↑↑			
2,5-dimethylpyrazine (M)	916	11.0	0.854	***	↑↑↑↑	N.F.		
benzaldehyde (M+D)	969	-	0.537	***	↑	−0.459	***	=
benzaldehyde (M)		11.5	0.537	***	↑	−0.471	***	=
benzaldehyde (D)		14.7	N.F.			−0.404	***	=
benzene acetaldehyde (M)	1049	12.6	0.669	***	↑↑	N.F.		
2-3H-furanone 5-ethylhydro (M+D)	1064	-	N.F.			−0.424	***	=
2-3H-furanone 5-ethylhydro (M)		11.9				−0.295	***	=
2-3H-furanone 5-ethylhydro (D)		15.3				−0.565	***	↓
*Ketones*								
1-hydroxypropan-2-one (M+D)	656	-	−0.414	***	=	N.F.		
1-hydroxypropan-2-one (M)		10.5	−0.664	***	↓↓			
1-hydroxypropan-2-one (D)		12.2	0.034	-	=			
pentan-2-one (D)	689	13.7	N.F.			−0.210	***	=
pentane-2,3-dione (M+D)	698	-	0.685	***	↑↑	N.F.		
pentane-2,3-dione (M)		12.2	0.368	***	=			
pentane-2,3-dione (D)		13.0	0.604	***	↑↑			
3-hydroxybutan-2-one (D)	716	13.4	0.158	-	=	N.F.		
hexan-2-one (M+D)	791	-	−0.191	-	=	0.056	-	=
hexan-2-one (M)		11.7	−0.191	-	=	N.F.		
hexan-2-one (D)		15.0	N.F.			0.056	-	=
heptan-2-one (M+D)	891	-	0.647	***	↑↑	−0.214	***	=
heptan-2-one (M)		12.6	0.647	***	↑↑	−0.285	***	=
heptan-2-one (D)		16.3	N.F.			−0.193	***	=
1-octen-3-one (M+D)	987	-	N.F.			−0.562	***	↓
1-octen-3-one (M)		12.8				−0.493	***	=
1-octen-3-one (D)		16.8				−0.585	***	↓
octan-2-one (M+D)	993	-	N.F.			−0.082	-	=
octan-2-one (M)		13.4				0.111	-	=
octan-2-one (D)		17.6				−0.181	***	=
nonan-2-one (M+D)	1098	-	N.F.			−0.205	***	=
nonan-2-one (M)		14.1				−0.146	**	=
nonan-2-one (D)		18.8				−0.242	***	=
*Acids*								
acetic acid (M+D)	601	-	0.138	-	=	−0.558	***	↓
acetic acid (M)		10.6	−0.290	*	=	N.F.		
acetic acid (D)		11.6	0.265	*	=	−0.558	***	↓
propanoic acid (M+D)	691	-	0.241	*	=	−0.127	*	=
propanoic acid (M)		11.2	0.183	-	=	−0.181	**	=
propanoic acid (D)		12.7	0.180	-	=	−0.039	-	=
isovaleric acid (M)	859	12.2	−0.631	***	↓↓	N.F.		
2-methylbutanoic acid (M)	865	12.0	−0.025	-	=	N.F.		
hexanoic acid (M)	997	13.0	−0.120	=	=	N.F.		

M: proton-bound monomer, D: proton-bound dimer. r: Spearman correlation coefficient. N.F.: the compound has not been detected in the stage. ↑↑↑↑: very strong positive/increasing trend (r > 0.8), ↑↑↑: strong positive/increasing trend (0.7 < r < 0.8), ↑↑: moderate positive/increasing trend (0.6 < r < 0.7), ↑: weak positive/increasing trend (0.5 < r < 0.6). ↓↓: moderate negative/decreasing trend (−0.7 < r < −0.6), ↓: weak negative/decreasing trend (−0.6 < r < −0.5). =: constant trend (−0.5 < r < 0.5). **** *p* < 0.0001, *** *p* < 0.001, ** *p* < 0.01, * *p* < 0.05, -: *p* > 0.05.

## Data Availability

The original contributions presented in the study are included in the article, further inquiries can be directed to the corresponding author.

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
