# Peer review of "Monitoring of the Dry-Curing Process in Iberian Ham Through the Evaluation of Fat Volatile Organic Compounds by Gas Chromatography–Ion Mobility Spectrometry and Non-Destructive Sampling"

_foods, 2024, doi:10.3390/foods14010049_

Round 1
Reviewer 1 Report (Previous Reviewer 2)
Comments and Suggestions for Authors
This manuscript has been improved. It is, however, strongly recommended to provide high resolution images.
Author Response
Comment 1: This manuscript has been improved. It is, however, strongly recommended to provide high resolution images.
We would like to thank the Reviewer for their time and effort towards improving our manuscript. Regarding the figures, we have improved their resolution in the revised manuscript. Moreover, we will provide them as separate files in the revised submission to ensure their quality.
Reviewer 2 Report (New Reviewer)
Comments and Suggestions for Authors
I have included some minor comments in the attached file. I often eat Iberian ham as it is available in my country, so I read the manuscript with great interest. Thanks to this work, I know that its consumption can be harmful to my health.

Author Response
General comment: I have included some minor comments in the attached file. I often eat Iberian ham as it is available in my country, so I read the manuscript with great interest. Thanks to this work, I know that its consumption can be harmful to my health.
We sincerely thank the Reviewer for their valuable contributions to improving our manuscript. All comments have been carefully considered, and we provide below a detailed response to each point raised.
With respect to the Reviewer’s comment regarding the harmful effects of Iberian ham on health, we would like to convey a message of reassurance about this valuable product. As can be read in our response to Comment No. 6 of the Reviewer, the toxic compound aniline, which was detected during the post-salting stage, was eliminated during the curing process and was not present at the drying-maturation stage. This demonstrates that this compound is absent in the final product, confirming that Iberian ham is a healthy product that poses no risk to consumer health
Comment 1: The year 1992 cannot be called ‘the last years’, it was 32 years ago. For science, this is a gulf.
Response 1: Thank you for pointing this out. We have, accordingly, changed this expression by “Over the years,”.
Comment 2: nonpolar
Response 2: The term has been changed in the revised manuscript.
Comment 3: As the ID of the column is 0.32 mm, a flow rate of 1 mL/min is too low. This flow rate would be suitable for a 0.25 mm column.
Response 3: GC-IMS is an exceptionally sensitive technique capable of detecting a wide range of signals, including both monomers and dimers of compounds, resulting in topographic maps with a high signal density. To enhance resolution in the lower region of the map, where signal overlapping is most pronounced, a flow rate of 1 mL/min was chosen for this method. Nevertheless, in light of the Reviewer's suggestion, this parameter will be carefully reevaluated.
Comment 4: Since you write below: alcohols, wstry then here you should be consistent and write aldehydes.
Response 4: The family “aldehydes” is indicated in the beginning of the table, in the previous page (page 5).
Comment 5: How many of these have been identified?
Response 5: A total of 36 compounds were identified in the post-salting stage from the 164 markers detected, while 27 compounds were identified in the drying-maturation stage from the 276 signals detected. This information is provided in lines 210–213 of the original manuscript.
Comment 6: From your results, I conclude that Iberian ham contains the carcinogen aniline and thus its consumption is not good for health.
Response 6: We understand the concern that the detection of toxic compounds in hams may raise. However, it is important to consider the high sensitivity of the GC-IMS technique, which allows for the detection of certain compounds even below their toxicity threshold. Since the presence of this compound has been reported in hams for other authors (reference below) It would be interesting to conduct a quantitative analysis in the future to determine the concentration levels at which this compound is present in hams and its evolution during the curing process.
Nonetheless, it should be noted that that this compound was only detected in the post-salting stage, and it was not found in the 38 samples evaluated in the drying-maturation stage. Therefore, we can affirm that the final product is completely safe for consumption.
del Pulgar, J. S., García, C., Reina, R., & Carrapiso, A. I. (2013). Study of the volatile compounds and odor-active compounds of dry-cured Iberian ham extracted by SPME. Food science and technology international, 19(3), 225-233.
This manuscript is a resubmission of an earlier submission. The following is a list of the peer review reports and author responses from that submission.
Round 1
Reviewer 1 Report
Comments and Suggestions for Authors
The study, entitled "Monitoring of dry-curing process in Iberian ham through the evaluation of fat volatile organic compounds by gas chromatography-ion mobility spectrometry and a non-destructive sampling ", analyzed key volatile components of Iberian ham at different curing stages. Overall, the manuscript is well written and generally easy to understand, with only a few missing pieces of information that need tweaking. Below, I offer some general and detailed comments:
1. Lines 108 to 115 should include more information about the dry curing conditions of the ham, such as changes in temperature and humidity from batch to batch.
2. Missing line numbers after page 11 of the manuscript.
3. On page 13, it is mentioned that ". Due to the seasonality of the slaughter of acorn-fed Iberian pigs (in January-February), drying-maturation stage coincides with the summer, taking advantage of the natural temperature to complete this stage in cellars." "However, earlier in the text, it is stated that the post-salting phase occurs in the summer and that high summer temperatures increased aldehyde levels. Is this a contradiction or a misstatement in the described phase?
4. While the manuscript provides many explanations for the different stages of volatile compounds, it does not discuss the potential interactions between different volatile compounds. I believe that exploring their interrelationships through model-based analysis can lead to a better understanding of the process of flavor formation.
5. I recommend further analysis of how these compounds specifically affect flavor characteristics, especially in terms of sensory changes. And it is recommended to cite more research on sensory evaluation as support.
Reviewer 2 Report
Comments and Suggestions for Authors
The purpose of this study was to monitor the dry-curing process of Iberian ham through the combination of a non-destructive collection methodology of the pieces and the volatile profile analysis by gas chromatography-ion mobility spectrometry. In the manuscript, authors only qualitatively analyzed the changes of volatile compounds types in Iberian ham at different stages using GC-IMS technology, the analysis method was single, and no quantitative analysis was carried out. This manuscript does not provide much more new knowledge and the work is lacking of novelty and importance. Meanwhile, the quality of the Figures in this manuscript are too poor.
1.Line 29: The “E” in (E)-2-octenal should be italic, please check the whole manuscript.
2. Line 37: The keywords are not accurate enough.
3. Line 56: The “Volatile Organic Compounds” should be revised to the “volatile organic compounds”.
4. Line 170-175: What do the intervals 0.6 < r < 0.75 and -0.75 < r <-0.6 respectively indicate?
5. In Table 1, the r value of the butanoic acid methyl ester is -0.13 at the dry maturation stage, which should be recognized as a weak negative correlation. However, it is represented by a straight arrow in Table 1.
6.The format of the references is not standardized.
Reviewer 3 Report
Comments and Suggestions for Authors
Comments to authors:
The paper is about using an established and non-destructive gas chromatography-ion mobility spectrometry method to monitor the dry-curing process of Iberian ham. Various VOCs were tentatively identified and discussed for samples taken during the post-salting stage and drying-maturation stage.
I find the paper of interest based on the method used and VOCs detection, however, the authors need to make some improvements (as pointed out under Comments to consider). I am not convinced about the reliability of the VOCs identification and that these markers are indeed key markers to use to monitor the process.
Comments to consider:
Title (L1-3)
The paper has a suitable title, but should it not be … Monitoring of [the] dry-curing process…
Abstract (L17-33)
L19: As a reader it is not clear to me what is meant with pieces (‘evaluation of pieces’ aroma’). I would advise the authors to describe this better in the abstract. The aim of the abstract does not align well with the aim at the end of the Introduction (L98-101). Overall, I would say that the abstract lacks crucial details and needs to be more specific.
Keywords (L36-37)
Consider including words which reflect terms relevant to the content, but not present in the title of the paper. Consider arranging alphabetically.
Introduction (L38-105)
L39: I would suggest to also mention the European countries for the production.
L40: Not clear what is meant by ‘elaboration’ procedure. Is this the correct term to use? Should it not be production procedure?
L44: I would probably be best to mention something about the slaughter and then the meat cuts subjected to dry-curing.
L99: Specifies that the ‘subcutaneous fat’ was measured. This was not clear in the abstract.
Overall, the Introduction is well written and structured. As reader, it is clear to me what has been done in the field and how this study will build on previous work. Also, the novelty is somewhat clear, but perhaps the authors could add a sentence to state what is novel about the study (‘what has not been done before or is different’).
Materials and methods (L106-175)
L108-110: I would recommend to provide more details about the hams and the animals they originated from. Were samples taken from individual animals? What is the experimental unit?
L113-115: The week sampling times are not clear. Perhaps the paper could benefit from a figure or table that shows these sampling points. Also, the authors needs to explain the logic behind the timepoint (week) selection.
L116-124: It would be good to mention that only the subcutaneous fat was sampled. Currently it is not clear from the text.
L128: Write out numbers in full at the start of the sentences (‘1 mL’).
L157-159: I suggest not to place the text in brackets.
L160-165: More information is needed to explain how the VOCs were identified and quantified (or lack of quantification). Were other databases used to verify the identification of the VOCs? Could the authors comment on the reliability of the identification.
Overall, the Materials and methods require additional information to allow another researcher to reproduce the results.
Results and discussion (L176-rest of line numbers missing)
L181-183: Based on the method of identification, how reliable is the statement that the study ‘constitutes the first description of (E,E)-2,4-hexadienal, 2,5-dimethylpirazine and 2-acetylpyrrole’?
Table 1 caption needs to include the number of samples used and compared in the table. All abbreviations used in the table also needs to be explained in the footer of the table.
Section 3.1: It would be of interest if the authors also explored the fingerprints using the VOCs that were not detected. Why was it only decided to focus on the identified VOCs? How sure are the authors that these VOCs are the key markers to assess the quality? What about the VOCs not identified?
The error bars in Figure 2 are rather large, would this then not make VOCs unreliable indicators?
Conclusions
I would suggest to specify that it was only the identified markers.